# Osteoclasts Differentiation from Murine RAW 264.7 Cells Stimulated by RANKL: Timing and Behavior

**DOI:** 10.3390/biology10020117

**Published:** 2021-02-04

**Authors:** Nadia Lampiasi, Roberta Russo, Igor Kireev, Olga Strelkova, Oxana Zhironkina, Francesca Zito

**Affiliations:** 1Consiglio Nazionale delle Ricerche, Istituto per la Ricerca e l’Innovazione Biomedica, Via Ugo La Malfa 153, 90146 Palermo, Italy; roberta.russo@irib.cnr.it (R.R.); francesca.zito@irib.cnr.it (F.Z.); 2Department of Electron Microscopy, Andrey N. Belozersky Institute of Physico-Chemical Biology, Lomonosov Moscow State University, 1-40 Leninskie Gory, 119234 Moscow, Russia; kireev@belozersky.msu.ru (I.K.); olga_strelkova@meei.harvard.edu (O.S.); zhironkina@belozersky.msu.ru (O.Z.)

**Keywords:** multinucleated cell, RANK localization, WGA-FITC labeling, NFATc1, p-p38, p-ERK, nuclear co-localization, migrasomes

## Abstract

**Simple Summary:**

The formation of multinucleated cells is critical for mature osteoclast generation. For this to happen, mononuclear pre-osteoclasts migrate in proximity to other pre-osteoclasts to fuse together. This finely regulated process is dependent on the “feeling” and “recognition” between neighboring cells. In this study, we focused on pre-osteoclast fusion timing and behavior, and highlighted changes in morphological and cytoskeletal organization during 4 days of osteoclast differentiation. For the first time, interesting cellular extensions have been described, with the presumed function to serve as migration mediators. The sub-cellular localization of key proteins correlated with the osteoclast maturation timing. In particular, for the first time, a relationship of mutual exclusion from the nuclei has been shown between two mitogen activated protein kinases and a master transcription factor. The different trends in the expression of some genes involved in the osteoclast fusion process were related to osteoclast differentiation timing. Although further investigation is needed, we are confident that this study will contribute to the understanding of the mechanisms regulating the initial processes of osteoclastogenesis, including migration and fusion, which in turn are of fundamental importance for the management of many bone-related diseases, such as osteoporosis, osteopetrosis and rheumatoid arthritis.

**Abstract:**

The development of multi-nucleated cells is critical for osteoclasts (OCs) maturation and function. Our objective was to extend knowledge on osteoclastogenesis, focusing on pre-OC fusion timing and behavior. RAW 264.7 cells, which is a murine monocyte-macrophage cell line, provide a valuable and widely used tool for in vitro studies on osteoclastogenesis mechanisms. Cells were treated with the receptor activator of nuclear factor κ-B ligand (RANKL) for 1–4 days and effects on cell morphology, cytoskeletal organization, protein distribution, and OC-specific gene expression examined by TEM, immunofluorescence, and qPCR. Multinucleated cells began to appear at two days of Receptor Activator of Nuclear factor κ-B Ligand (RANKL) stimulation, increasing in number and size in the following days, associated with morphological and cytoskeletal organization changes. Interesting cellular extensions were observed in three days within cells labeled with wheat germ agglutinin (WGA)-Fluorescein isothiocyanate (FITC). The membrane, cytoplasmic, or nuclear distribution of RANK, TRAF6, p-p38, pERK1/2, and NFATc1, respectively, was related to OCs maturation timing. The gene expression for transcription factors regulating osteoclastogenesis (*NFATc1*, *c-fos*, *RelA*, *MITF*), molecules involved in RANKL-signaling transduction (*TRAF6*), cytoskeleton regulation (*RhoA*), fusion (*DC-STAMP*), migration (*MMP9*), and OC-specific enzymes (*TRAP, CtsK*), showed different trends related to OC differentiation timing. Our findings provide an integrated view on the morphological and molecular changes occurring during RANKL stimulation of RAW 264.7 cells, which are important to better understand the OCs’ maturation processes.

## 1. Introduction

Osteoclasts (OCs) are multi-nucleated cells, originating from the monocyte/macrophage lineage of hematopoietic stem cells, playing a fundamental role in skeletal development, bone remodeling, and fracture healing [1].

Under stimulation by appropriate cytokines, OCs precursors undertake osteoclastogenesis, which is a very complex and sophisticated differentiation process, involving pre-OCs proliferation, their migration, cell–cell adhesion, and fusion, eventually leading to the formation of mature multinucleated OCs [2]. The two essential cytokines are the Macrophage-Colony Stimulating Factor (M-CSF), stimulating migration and supporting pre-OCs proliferation, and the Receptor Activator of Nuclear factor κ-B Ligand (RANKL), which is crucial for the maturation of pre-OCs in multinucleated OCs. The events downstream the binding to their corresponding receptors, c-fms and RANK, respectively, have been widely described [2]. The RANKL signaling pathway involves the recruitment of adapter proteins, i.e., TNF receptor associated factors (TRAFs), and leads to the activation of mitogen-activated protein kinases (MAPKs) and transcription factors, such as AP-1, NFATc2, and then NFATc1 [2].

MAPKs are important molecules that transduce external signals into internal cellular responses. It has been demonstrated that p38, c-Jun N-terminal kinase (JNK), and extracellular signal-regulated kinase (ERK) transient phosphorylation promotes the expression of various OC marker genes including NFATc1 [3,4,5]. More recently, the axis RANKL-RANK-TRAF6-p38 MAP kinase was related to the onset of osteoclastogenesis [6] and RANKL-RANK-MEK-ERK-NFATc1 axis was related to migration-fusion in RAW 264.7 cells [7]. NFATc1 has been shown to play a role as a master regulator of osteoclastogenesis, regulating the expression of numerous osteoclast-specific molecules involved in fusion, in differentiation, and maturation of multi-nucleated OCs, and in bone remodeling [8]. Our previous results showed that the expression of osteoclast hallmarks depends on the induction of NFATc1 and ingenuity pathway analysis (IPA) correlated NFATc1 with the activation of ERK and p-38 MAPKs [9].

In this study, we analyzed timing and behavior of OCs during their in vitro differentiation. We used the RAW 264.7 cells, which are a mouse macrophage cell line with the capacity to differentiate into OCs after RANKL stimulation. We investigated osteoclastogenesis at a morphological level, studying OCs phenotypes by electron microscopy and proteins localization, distribution by immunofluorescence analyses at a molecular level, and studying the mRNA expression of a panel of genes by qPCR analysis.

## 2. Materials and Methods

### 2.1. Cell Culture and Osteoclastogenesis In Vitro

The murine RAW 264.7 cell line was purchased from the American Type Culture Collection (ATCC) (Manassas, VA, USA). Cells were grown in Dulbecco modified Eagle’s medium (DMEM, Gibco, NY, USA), with 10% heat-inactivated fetal bovine serum (FBS, Sigma-Aldrich, St. Louis, MO, USA,), 100-U/mL penicillin, and 100-µg/mL streptomycin. To induce OC differentiation, cells were suspended in alpha-minimal essential medium (α-MEM Gibco, Grand Island, NY, USA) with 10% heat-inactivated fetal bovine serum (FBS, Sigma-Aldrich, St. Louis, MO, USA), 100-U/mL penicillin, and 100-µg/mL streptomycin, with the addition of 50-ng/mL RANKL (Peprotech, Rocky Hill, CT, USA). RAW 264.7 differentiation toward multinucleated OCs has been followed in vitro over four days, considering as day 0, the day the cytokine RANKL was added, which was replaced every two days to maintain the signal.

For the experiments of OC differentiation on extracellular matrix (ECM) proteins, 8-well chamber slides were pre-coated with 20 µg/mL of fibronectin (FN, Sigma-Aldrich, St. Louis, MO, USA) or recombinant osteopontin (OPN, Sigma-Aldrich, St. Louis, MO, USA) in phosphate buffered saline (PBS), overnight at 4 °C [10]. To increase the amount of proteins physically adsorbed onto the wells, an additional incubation was performed for 1 h at 37 °C. The wells were then blocked using 1% bovine serum albumin (BSA) in PBS for 1 h at 37 °C in a CO_2_ incubator. Finally, the wells were washed with PBS and, to prevent denaturing of the proteins, left in PBS until cells were plated. RAW 264.7 cells were plated at 1 × 10^4^ cells/well in 8-well chamber slides overnight.

### 2.2. Electron Microscopy Analysis

For electron microscopy experiments, cells (2.5 × 10^5^/well/6 wells plate) were grown on coverslips, fixed in 2.5% glutaraldehyde in 0.1 M Sorenson’s buffer (pH 7.4) for 2 h with subsequent post-fixation in 1% OsO4 and embedded in Epon (Sigma-Aldrich, St. Louis, MO, USA). Serial ultrathin sections (90 nm) were cut with Leica Ultracut-E ultramicrotome (Leica Microsystems, Wetzlar, Germany), stained with 2% aqueous uranylacetate and lead citrate, and observed with JEM-1400 TEM microscope operated at 100 kV (JEOL, Tokyo, Japan).

### 2.3. Tartrate-Resistant Acid Phosphatase (TRAP) Staining

RAW 264.7 cells (2.5 × 10^5^/well/plate 6 wells) were cultured for four days with RANKL (50 ng/mL) as previously described and then fixed with 4% paraformaldehyde in phosphate buffered saline (PBS) for 10 min at room temperature. TRAP staining was carried out in accordance with the manufacturer’s instruction (Sigma Aldrich, St. Louis, MO, USA). Cells were observed under light microscopy (OLYMPUS CKX31, Olympus, Tokyo, Japan), and those containing more than three nuclei were considered OCs.

### 2.4. Immunofluorescence Analysis

For indirect immunofluorescence experiments, RAW 264.7 cells (2.5 × 10^5^/well/plate 6 wells) were grown on coverslips for various time periods with or without RANKL (50 ng/mL). The analysis of the cell surface localization of RANK was performed according to Fiorino et al. [11] with few modifications. Cells were washed with cold TBS/1 mM Ca^2+^ and incubated with anti-RANK antibody (1:200, Santa Cruz, CA, USA) for 20 min at 4 °C. Then, cells were fixed with 4% paraformaldehyde (PFA, Sigma-Aldrich, St. Louis, MO, USA), blocked with 5% foetal calf serum, and incubated with secondary antibody Alexa Fluor-594 rabbit anti-mouse (1:500, Invitrogen Molecular Probes, Life Technologies, Carlsbad, CA, USA) for 1 h at room temperature in a dark, humidified chamber. For the analysis of cytoplasmic antigens, cells were fixed with 4% PFA in PBS for 15 min, permeabilized with 0.5% Triton X-100 (Sigma)/PBS for 10 min, and blocked with 1% BSA (Sigma)/0.1% Triton X-100/PBS for 30 min. Alexa Fluor-488 phalloidin (1:250, Invitrogen Molecular Probes, Carlsbad, CA, USA), WGA-FITC (1:400, Sigma-Aldrich, St. Louis, MO, USA), and antibodies anti α-tubulin (1:500, Sigma-Aldrich, St. Louis, MO, USA), RANK (1:20, Santa Cruz, CA, USA), TRAF6 (1:20, Santa Cruz, CA, USA), p-p38 (1:20, Cell Signaling Technology, Beverly, MA, USA), p-ERK (1:20, Cell Signaling Technology, Beverly, MA, USA), and NFATc1 (1:20, Santa Cruz, CA, USA) were incubated for 2 h at room temperature in a dark, humidified chamber and secondary antibodies Alexa Fluor-488 rabbit anti-mouse (1:200, Invitrogen Molecular Probes, Carlsbad, CA, USA) and Alexa Fluor-594 goat anti-rabbit (1:200, Invitrogen Molecular Probes, Carlsbad, CA, USA) were incubated for 1 h at room temperature. All the coverslips were mounted on glass slides with mounting medium containing 4′,6-diamidino-2-phenylindole (DAPI) (Invitrogen Molecular Probes, Carlsbad, CA, USA) and observed under an Axioskop-2 plus microscope (Zeiss, Oberkochen, Germany), equipped for epifluorescence. Images were recorded using a digital camera system, then cropped, and grouped. The contrast and brightness were adjusted using Photoshop CS2 (Adobe Systems, San Jose, CA, USA). Samples labeled with p-ERK were observed under a Nikon Eclipse 80i microscope (Nikon, Kawasaki, Kanagaw, Japan) equipped for epifluorescence and recorded by a digital camera system (40×). The percentage of cells showing the nuclear localization of various proteins was calculated as the ratio of nuclei labeled with specific antibodies to the total number of nuclei labeled with DAPI (Molecular Probes, Eugene, OR, USA), counted using the ImageJ software (https://imagej.nih.gov/ij/, 1997–2018).

### 2.5. RNA Extraction and cDNA Synthesis

RAW 264.7 cells were cultured in a 6-well plate (1 × 10^6^ cells/well) overnight. Cells were treated or untreated (ctrl) with 50 ng/mL RANKL for 1, 2, 3, and 4 days. At each stimulation time, cells (treated and untreated) were detached from the wells and washed once with PBS. Total RNA was isolated using the “GenElute Mammalian Total RNA Miniprep Kit” (Sigma-Aldrich, St. Louis, MO, USA) and quantified by using the bio photometer (Eppendorf, Hamburg, Germany). Total RNA (3 μg) was converted to cDNA using the SuperScript Vilo (Invitrogen Molecular Probes, Carlsbad, CA, USA).

### 2.6. qPCR

Quantification of gene expression was performed using the StepOnePlus real-time PCR as described in the manufacturer’s manual (Applied Biosystems, Grand Island, NY, USA) with SYBR Green chemistry, and the Comparative Threshold Cycle Method [12]. The qPCR was run as follows: 1× cycle, denaturing at 95 °C for 10 min for DNA polymerase activation, and 38 cycles, melting at 95 °C for 15 s and annealing/extension at 60 °C for 60 s. QPCR was then performed using primer by Qiagen (Germantown, MD, USA): QT001676692 (NFATc1), QT00166663 (TRAF6), QT00108815 (MMP9), QT00131313 (MITF1), QT00131012 (TRAP), QT01047032 (DC-STAMP), QT02589489 (CTSK2), QT00149415 (RelA), QT00197568 (RhoA), QT01167257 (OSCAR), QT009399 (RANK), QT01658692 (GAPDH), FOS F: 5′CACTCCAAGCGGAGACAGAT3′, and R: 5′TCGGTGGGCTGCCAAAATAA3′. The threshold cycle (CT) values were calculated against the housekeeping gene GAPDH, whose expression was not affected by the experimental conditions. The expression levels of the analyzed genes are presented as relative values of the cells treated with RANKL compared to the cells not treated with RANKL (ctrl). Each day has its appropriate control on the same day (assumed with a value of 1). At least three distinct biological samples were examined for each gene and treatment. Data are expressed as mean ± S.D.

### 2.7. Statistical Analysis

Data are expressed as mean ± S.D. from at least three experiments and statistical analyses were performed by Student’s *T* test. *p* < 0.05 was considered to indicate a statistically significant difference.

## 3. Results

### 3.1. Timing of In Vitro Osteoclast Differentiation

To follow osteoclastogenesis in vitro, RAW 264.7 cells were stimulated with RANKL for a period of four days and observed for the formation of multinucleated cells (Figure 1A–E). In the absence of RANKL stimulation (resting condition), cells were mono-nucleated and mainly showed a rounded morphology (Figure 1A, RANKL^−^). Bipolar cells with long filopodia were observed in the RANKL-stimulated cell population at day 1 (RANKL^+^), together with very few bi-nucleated cells (Figure 1B). Fusion is an event that usually begins between few cells (less than 10%) (Appendix A). From the two days onwards, the number of cell–cell fusion events increased, leading to about 20–25% of cells containing more than 2 nuclei, with the formation of large-sized multi-nucleated cells (about 30% of cells with more than four nuclei), as observed at 3 and 4 days (Figure 1A). Since the fusion process was not exactly synchronous, at any time point past RANKL addition, especially at 2 and 3 days, it was possible to observe intermediate stages of the cell fusion process within the same population of RANKL-stimulated cells. The day 2 sample was selected for TEM analysis because the fusion process at this time point is very active, with few large multinucleate cells already present in a sufficient number (Figure 1E), together with cells in the phases of recognition/adhesion as well as fusion (Figure 1B–D). The recognition between mononuclear cells is featured by two approaching cells (see asterisks in Figure 1(B1,B2)), whereas the adhesion was manifested through a wide contact area of the membranes of two close cells (see arrowheads in Figure 1(B1)), which sometimes appeared very jagged and interconnected (see magnification in Figure 1(C1)). The actual fusion occurs through a “fusopode bridge” where small membrane discontinuities (fusion pores) occur, leading to cytoplasm mixing, which can be appreciated by observing cytoplasmic continuity between the two cells (see arrow in Figure 1D). The presence of active multinucleated OCs was further confirmed by the positive staining of 4 days RANKL-stimulated cells for tartrate-resistant acid phosphatase (TRAP), which is an osteoclast-associated enzyme that is a specific and sensitive indicator of bone resorption (Figure 1F).

More in-depth observations on the approaching cells during their recognition and fusion can be obtained by labeling them with wheat germ agglutinin (WGA), which is a lectin that recognizes N-acetylglucosamine and terminal neuraminic acid [13] and has been successfully used to stain cells for the detection of the plasma membrane of syncytia [14]. Within the population of 3 days RANKL-stimulated cells, numerous morphologies can be observed (Figure 2), depending if they are mononucleated or multinucleated cells. The multinucleated cell edges were very irregular, showing numerous cellular extensions, either short filopodia (white arrow in Figure 2(A3)) and longer ones (Figure 2A,B,D). Some cells displayed typical dendrite-like extensions with numerous filopodia and lamellipodia (see white arrowhead in Figure 2(A3)). Furthermore, we sometimes observed interesting structures, especially in mononuclear cells approaching others, i.e., some sort of vesicles along a number of filopodia (see white arrows in Figure 2(B3,D2)) or apparently attached to the glass substrate with no connections to the cells (Figure 2(C2,C3)). The vesicles were variable in size and in amount per filopodia.

### 3.2. Cytoskeleton Organization during In Vitro OC Differentiation

The organization of the cytoskeleton is a critical aspect to be taken into consideration for the OC differentiation. After 1 and 2 days of RANKL stimulation, some podosome arrangements can be observed in mononuclear and binuclear cells, mainly in the form of small F-actin clusters (see white arrows in Figure 3). The typical changes in podosomes organization were observed at 4 days after RANKL addition, when actin belts appeared at the edges of multi-nucleated OCs. Cells cultured without RANKL for the same time did not fuse and mononuclear cells did not show any particular organization of actin and tubulin molecules (4 days/RANKL^−^) (Figure 3).

It is known that the OCs’ morphology is modulated by the extracellular matrix (ECM) proteins on which cells adhere [10]. Here, we focused on fibronectin (FN) and osteopontin (OPN) effects on early stages of OC differentiation, i.e., 1 and 2 days after RANKL addition, during which pre-OCs are mainly involved in migration. About 7–8% of mononuclear cells grown on FN and OPN had a bipolar morphology with very long and spiny filopodia, at 1 day after RANKL addition (Figure 4). Some dendrite-like extensions were observed in about 25% of the cells cultured on OPN (see white arrows in Figure 4). Moreover, about 24% of cells cultured on OPN showed actin rings that were never observed in cells grown for 1 day on glass after RANKL addition (compare Figure 3 with Figure 4). The presence of numerous actin rings was also observed in 2-day cells, mononucleated and multinucleated, both on FN and OPN (Figure 4).

### 3.3. Sub-Cellular Localization of RANK and NFATc1 during In Vitro OC Differentiation

RANK, the RANKL receptor, and NFATc1, the master transcription factor for osteoclastogenesis, are known to have a critical role during OC differentiation. Here, their localization in RAW 264.7 cells was analyzed under a resting condition and for 4 days after RANKL stimulation, by immunofluorescence analysis (Figure 5). In the absence of stimulus, RANK was present in almost all cells, where it was localized primarily on the cell membrane (see an example in red framed 1.5× magnification in 1 day/RANKL^−^ in Figure 5A). From 1 day onwards in the presence of RANKL, a reduced number of mononuclear cells continued to show RANK on their membrane, compared to RANKL^−^ control cells (see an example in red framed 1.5× magnification in 2 day/RANKL^+^ in Figure 5A). On the contrary, all the multinucleated cells, observed from 2 to 4 days, did not show RANK on their surface (see white arrows in Figure 5A), which was widespread in their cytoplasm or completely absent.

The NFATc1 protein is mainly localized in the cytoplasm of unstimulated cells (1 day/RANKL^−^ in Figure 5B) as well as in the cells stimulated for 1 day with RANKL (1 day/RANKL^+^ in Figure 5B). After 2 days of RANKL stimulation, along with the appearance of multinucleated cells, the translocation of NFATc1 to the nucleus became evident in several mononuclear and some multinucleated cells (see white framed 1.5× magnification within 2D/RANKL^+^ in Figure 5B), while it maintained a low level and diffused cytoplasmic distribution in the majority of the cells (see yellow framed 1.5× magnification within 2 days/RANKL^+^ in Figure 5B). The NFATc1 protein maintained a nuclear localization even during late OC differentiation. However, in the large multinucleated cells observed at 3 and 4 days, not all the nuclei were labeled with the same intensity (compare the white-framed and yellow-framed magnifications in 3 days/RANKL^+^ in Figure 5B).

### 3.4. Sub-Cellular Localization of MAPKs Signaling Intermediates during In Vitro OC Differentiation

The co-localization of some intermediate molecules in the RANKL-RANK pathway, i.e., TRAF6/p-p38 MAPK (Figure 6), NFATc1/p-p38 MAPK (Figure 7), and NFATc1/p-ERK (Figure 8), was analyzed in RAW 264.7 cells under resting condition and during RANKL stimulation (1, 2, 3 days) by immunofluorescence analysis. The choice to follow these days is based on the observation that these are the crucial days to observe a heterogeneous population of cells in different phases of OC differentiation, i.e., mononucleated and small and large multi-nucleated cells. In 2 days/RANKL^−^ and 2 days/RANKL^+^ cells, TRAF6, which is an adapter protein involved in the transduction of RANKL-mediated signaling, was always localized in the cytoplasm. However, some cells showed a low signal intensity (see yellow arrows in Figure 6), while the majority of them showed high signal (Figure 6). In multi-nucleated cells, TRAF6 was mainly widespread around the nuclei and at the membrane surrounding the vacuoles (see turquoise arrow in 3 days/RANKL^+^, Figure 6). On the contrary, the phosphorylated form of p38 MAPK (p-p38) was always localized within the nuclei of almost 100% of mononucleated and multinucleated cells, both in resting and RANKL-stimulated cells. However, in some nuclei, the signal intensity was much stronger than in others (see white arrows in Figure 6). There appears to be a co-localization of stronger p-p38 and TRAF6 signals.

The nuclear localization of NFATc1 (about 8.4%, 19.5%, and 24% at 1, 2, and 3 days, respectively) (see white frames 1 and 2 in 1.5× magnifications in Figure 7) was always associated with a less intense signal of nuclear p-p38, both in mononucleated and multinucleated cells, while the high intense signal of p-p38 was associated with cytoplasmic localization of NFATc1 (see white frames 3 and 4 in 1.5× magnifications in Figure 7).

Differently from p-p38 that was always localized in the nuclei, p-ERK was localized only in a small number of nuclei of RANKL^+^ cells, i.e., about 9.2% at 1 day and 1.5% at 2 and 3 days (see white and yellow frames of 1.5× magnifications in Figure 8). Moreover, about 67% of NFATc1 positive cells showed a nuclear co-localization with p-ERK at 1 day (see white frames of 1.5× magnifications in Figure 8), while this percentage decreased to about 6.5% on both days 2 and 3.

### 3.5. Gene Expression Analysis during In Vitro OC Differentiation

To study OC differentiation timing at a molecular level, we examined the gene expression profiles of different categories of genes (Figure 9). Among them, genes code for transcription factors (TFs) regulating osteoclastogenesis (Fos, MITF, RelA, NFATc1), for RANK and TRAF6, for a collagen receptor co-stimulator of osteoclastogenesis (OSCAR), and for molecules involved in cytoskeleton regulation (RhoA), cell-cell fusion (DC-STAMP), cell migration (MMP9), and effector proteins with enzymatic activities specific for multinucleated OCs (TRAP, CtsK).

Among the TFs analyzed, NFATc1 was highly upregulated, starting from 1 day after RANKL stimulation (29-fold increase), followed by a slight decrease for 2 and 3 days (21-fold and 14-fold increase, respectively) and a further rise on the fourth day (24-fold increase) (Figure 9A). The expression of Fos was not detectable during the early phases of differentiation (1 and 2 days), but it was induced from the third day (4-fold increase), reaching a 12-fold increase by day 4 (Figure 9A). RelA showed no significant variations in its expression levels during the entire differentiation process, while MITF expression was slightly induced only during day 3 (two-fold increase) (Figure 9A). TRAF6 expression levels showed no significant variations until day 4, when a 2.5-fold increase was observed (Figure 9B). RANK mRNA levels increased significantly during the OC differentiation time (Figure 9B). In particular, approximately a two-fold increase was observed at day 1 and 2, while a five-fold increase was observed for 3 and 4 days (Figure 9B). Among the proteins regulating cell migration and fusion, we found a fluctuating expression for RhoA, which remained at the basal levels (1 and 3 days), or was downregulated after 2 and 4 days (Figure 9C). On the contrary, DC-STAMP expression was highly upregulated starting from day 1 day and during days 2 and 3, with an average four-fold increase (4-fold ± 0.31), followed by a striking inhibition after four days (Figure 9C). MMP9 expression showed an induction from 1 day of differentiation (two-fold increase), which further increased at 3 (4-fold ± 0.46) and 4 days (3-fold ± 0.63) (Figure 9C). Among the proteins specific for multinucleated OCs, TRAP mRNA was the one highly up-regulated starting from 1 day (14-fold increase), followed by a further significant increase at 2 days (58-fold), a drastic decline at 3 days (3-fold ± 0.51), and even an inhibition at 4 days (Figure 9D). The expression of OSCAR was highly up-regulated at 2 days (43-fold increase), followed by a decrease from 3 days on (5-fold ± 2.5) (Figure 9D), while the expression of CtsK had a strong increase only at 3 days (5-fold ± 0.89), followed by a dramatic inhibition at 4 days (Figure 9D).

## 4. Discussion

In the present study, we investigated the timing and behavior of OCs during their in vitro differentiation, evaluating morphological changes, cytoskeleton organization, and sub-cellular distribution of some proteins as well as the expression of a panel of OC-specific genes. Utilizing the RAW 264.7 cells, which come from a murine monocyte-macrophage cell line, we were able to study the sequential events leading to OC differentiation induced only by RANKL. Our results clarify several points in the OC differentiation process that have not been highlighted so far. First, we show that morphological changes and cytoskeleton organization are related to the timing of RANKL stimulation as well as to the substrate on which RANKL-stimulated cells adhere. Pre-OCs and mature OCs are very dynamic cells, which alternate among migration times, cell-cell contacts, fusion, and resorption, resulting in different morphologies and a series of cytoskeleton rearrangements [15]. By our results, it appears evident that the first day after RANKL addition is characterized by pre-OCs search for a partner to fuse with, as shown by the presence of numerous cells with bipolar morphology. The filopodia emitted by the pre-OCs were of different sizes and shapes, depending on the substrate they adhered to, which might reflect the involvement of different adhesion molecules [2]. Furthermore, the appearance of numerous actin rings during early times of OC differentiation (1 or 2 days of RANKL stimulation) on FN or OPN, but not on a glass substrate, confirms the role played by the ECM in modulating OCs morphology [10].

The second interesting point are the vesicles observed along the filopodia of 3 days/RANKL^+^ cells, or sometimes disconnected by cells, shown by the WGA-FITC labeling. To our knowledge, this type of structure has never been described in the OCs and it is difficult to explain its presence and possible function during osteoclastogenesis. However, these vesicles resemble “migrasomes,” i.e., cellular organelles recently described, which form on retraction fibers of migrating cells as large vesicle-like structures [16]. The migrasome role has not yet been demonstrated, but it is proposed to provide spatio-temporal chemical information for cell-cell communication during cell migration [17]. Recently, Hu et al. [18] showed the release of a large number of cholesterol-rich particles by macrophages during their locomotion. The particles were fragments of plasma membrane, released during projection and retraction of filopodia and lamellipodia, which remained anchored to the substrate [18]. In this case, the particles were suggested to contribute to cholesterol transport.

The sub-cellular distribution of molecules involved in the RANKL-RANK pathway related to the OC differentiation timing has not been previously described. Studying RANK distribution during RANKL stimulation, its surface localization in almost all unstimulated cells as well as in mononuclear RANKL stimulated cells was expected, suggesting that they likely corresponded to OC-precursors, which are a pool of cells ready to undergo OC differentiation since they are able to rapidly respond to RANKL [19]. However, it is not clear the reason why, during RANKL treatment timing, the number of mononuclear cells expressing RANK on the surface decreased. The finding that a small percentage of RAW 264.7 cells are RANK^+^ pre-OCs is related to the observation that a little amount of cells undergo fusion after RANKL stimulation (Appendix A). Levaot et al. [20] assessed that OCs fusion began by a small subset of RANKL-stimulated pre-OCs (about 2.4%, but always below 10%), termed “fusion founders,” which fuse with other founders or with non-stimulated progenitors, termed “fusion followers.” The authors described an initial pre-fusion phase, during which a founder and a follower cell pair together, developing cytoplasmic communication. However, both of them maintain their own morphology for a while before the fusion event becomes apparent [20] (see Figure 1B,D and Appendix A in this study). Thus, an emerging concept concerns the heterogeneity between fusion partners, which show differences in both biological and molecular characteristics. It seems that fusion preferably occurs between partners of dissimilar motility and differentiation stages, i.e., a more mature and immobile OC prefers to fuse with a less mature and mobile pre-OC, which could be a strategy to control fusion in vivo [21].

To study the intracellular signaling pathways involved in pre-OC differentiation, we focused on the 1, 2, and 3 days after RANKL addition, as, during these days, multinucleated OCs appear and increase in number and size, leading to a heterogeneous population of cells in different phases of OC differentiation, i.e., mononucleated and small and large multi-nucleated cells (see schematic representation of morphologies in Figure 10). The binding of RANKL to its RANK receptor on the surface of the pre-OCs leads to a series of signaling cascades, involving, among others, the p38 and ERK MAPK signaling pathways [22]. Since RANK lacks an intrinsic enzymatic activity, the intracellular transduction of the RANKL signal is performed by recruiting adapter molecules, such as TRAF6, which has been shown to activate p38 and JNK MAPK during OC differentiation [6]. The cytoplasmic distribution of TRAF6 in RAW 264.7 cells at different phases of OC differentiation is a validation of its crucial role in transducing RANKL–RANK signaling. The nuclear co-localization of the stronger TRAF6 (cytoplasm) and p-p38 (nucleus) signals in RAW 264.7 cells is in agreement with the requirement of TRAF6 recruitment to the cytoplasmic tail of RANK for the RANKL-dependent p38 activation in bone marrow-derived mononuclear pre-OCs [6].

The nuclear localization of the active form of p38 (p-p38) in almost all RAW 264.7 cells regardless of the culture conditions was surprising. Usually, p38 MAPK is localized in the cytoplasm of resting cells and translocates to the nucleus after extracellular stimulation [23]. The ability of p38 to shuttle to the nucleus and back after extracellular stimulation is well known, even though it does not contain nuclear localization or nuclear export signals [24]. Nevertheless, in a few cases, p38 has been detected in the nucleus of resting cells, thus, suggesting its dual ability to phosphorylate substrates in the cytoplasm as well as in the nucleus [24]. NFATc1 is one of the targets of p38 MAPK. However, contrasting data have been reported concerning the relationship between them. Among various reports, Matsumoto and colleagues [3] demonstrated that p-p38 directly phosphorylates NFATc1 in murine bone marrow cells, re-locating it from the cytoplasm into the nuclei, while p-p38 promotes, albeit moderately, NFATc1 nuclear expulsion in T cells [25]. Our observation that the nuclear localization of NFATc1 was always associated with a less intense signal of nuclear p-p38, both in mononucleated and multinucleated OCs, suggests that they are mutually exclusive in nuclei after RANKL stimulation, supporting the idea that p-p-38 might contribute to NFATc1 nuclear export in RAW 264.7 cells. The relationship between p-p38 and NFATc1 in RAW 264.7 cells deserves further study (see schematic representation in Figure 10).

We recently deepened the relationship between p-ERK and NFATc1, showing that the long-lasting ERK activity depends on RANKL-dependent NFATc1 induction [7]. A significant decrease of ERK phosphorylation was observed after 24 h NFATc1 impairment (siRNA silencing) and, conversely, a decreased expression of NFATc1 when ERK phosphorylation was affected by FR180204 inhibitor treatment for 24 h [7]. The present results on p-ERK localization are in agreement with the previous ones, i.e., the major number of p-ERK^+^ cells was observed at 1 day after RANKL addition, while this amount severely decreased after 2 and 3 days. Furthermore, the major number of cells showing p-ERK/NFATc1 co-localization was observed on day 1, suggesting a strict relationship between them as a consequence of RANKL stimulation (see schematic representation in Figure 10).

During OC differentiation, NFATc1 translocates to the nucleus and activates both OC-specific genes and its own transcription [26]. In this study, we noticed a selective distribution of the NFATc1 protein in certain nuclei within individual large multi-nucleated OCs, shown by an unequal labeling intensity of the various NFATc1+ nuclei or even by its complete absence in others. This observation suggests that only a selection of nuclei within multinucleated OCs are transcriptionally active, even with different levels of activity, as already shown by Youn et al. [27] in RAW 264.7 cells.

The remarkable upregulation of *NFATc1* mRNA observed in RAW 264.7 cells since the 1st day after RANKL stimulation is a hallmark aspect of osteoclastogenesis. By our previous qPCR and bioinformatic analyses, we found that NFATc1 significantly influences the expression of 31 genes, which are differently involved in osteoclastogenesis [9]. A variety of data in literature support the idea that the initial induction of NFATc1 protein depends upon the cooperation of NFATc2 and NF-κB (in particular, components p50 and p65, named *RelA*), both recruited to the NFATc1 promoter at the very early phase of OC differentiation, i.e., 1 h after RANKL stimulation. Both NFATc2 and NF-κB are present in the cytoplasm of unstimulated cells, to be readily available to enter the nucleus and activate target genes soon after RANKL stimulation. *NFATc1* and the osteoclast hallmarks were not transcribed after 1 h of RANKL-induction whereas they are expressed after 24 h [7]. Consistent with these data are the levels of *RelA* mRNA measured in our cellular system, which are present in resting cells as well as in the first day and during all OC differentiation (Figure 9). On the contrary, from day 1 onwards after RANKL stimulation, c-Fos and NFATc1 itself are recruited to the NFATc1 promoter, as shown by ChIP analysis, and this occupancy persists toward terminal differentiation of OCs [28] in agreement with our results (Figure 9). MITF has been suggested to function as the most distal factor in osteoclastogenesis, being an NFATc1 signal modulator [29]. In particular, it has been shown that MITF can amplify the NFATc1-dependent expression of many downstream OC-specific genes, including *CtsK*, *OSCAR*, and *TRAP* [29] in agreement with our results showing *MITF* expression significantly increased at 3 days after RANKL-induction.

Cell migration is driven by the activity of the cytoskeleton, which undergoes rapid changes in its organization to accomplish cycles of movement and attachment during pre- and OCs migration. Our results showing the constant and significant increase in the expression of *MMP9* induced by RANKL from the first day after its addition are coherent with the known migrating activity of pre- and mature OCs under normal conditions. DC-STAMP is one of fusion-mediating molecules directly regulated by NFATc1, since the gene possesses a binding site for this TF in its promoter region [30]. DC-STAMP is a transmembrane molecule, which is found on the surface of most of the unstimulated pre-OCs, while it moves from the cell surface to become mainly cytoplasmic after RANKL stimulation, suggesting an event of internalization when cells begin to differentiate toward OCs [30]. Given that the expression of *DC-STAMP* mRNA is upregulated from the 1st day of RANKL stimulation, we can hypothesize that RANKL-induced DC-STAMP expression is necessary to restore it on the surface of differentiating OCs as a response to its previous internalization, which is in agreement with data reported in literature [30]. The mRNA expression levels of *DC-STAMP* and *NFATc1* peaked at the same time points after RANKL stimulation and NFATc1-knockdown significantly inhibited *DC-STAMP* expression [9]. Several results suggest that there is a mutual regulation between DC-STAMP and NFATc1 at both gene and protein expression levels [30]. The decreased expression of *NFATc1* has been recently described in DC-STAMP−/− cells [31].

OSCAR is an osteoclast-specific immuno-receptor, which acts as a co-stimulatory signal required for RANKL-mediated activation of NFATc1 and, in turn, it is a direct target of NFATc1, implying the presence of a positive feedback circuit OSCAR-NFATc1-OSCAR [8]. Recently, OSCAR has been shown to bind to specific motifs within fibrillar collagens, suggesting that ECM plays an active role in the OSCAR-mediated regulation of osteoclastogenesis [32]. It is generally acknowledged that the ECM molecules significantly affect cell behavior and influence the activities of several intracellular signaling pathways [33].

## 5. Conclusions

This study clarifies some aspects of the cellular differentiation of OCs. In particular, some relationships between MAPKs and proteins are important for the formation of mature OCs. Among them are the membrane receptor RANK, the master regulator of osteoclastogenesis NFATc1, and the adapter protein TRAF6. It also highlights, for the first time, a relationship of mutual exclusion from the nuclei between p-p38 and NFATc1 or p-ERK and NFATc1. A summary view of these relationships is schematically shown in Figure 10. Another important aspect for the formation of mature OCs is the presence of multinucleated cells. Mononuclear pre-OCs migrate in proximity to other pre-OCs to fuse. This finely regulated process depends on the “feeling” and “recognizing” of neighboring cells. For the first time in this study, structures similar to the “migrasomes,” already described in cancer and stem cells, are described. The proposed function for migrasomes is to serve as migration mediators, left behind by cells that migrate first, to point the way to other cells. This function could also be important for pre-OCs that must migrate to fuse. Little is known about migrasomes and further studies are required to understand the mechanisms that regulate migration-fusion.

## Figures and Tables

**Figure 1 biology-10-00117-f001:**
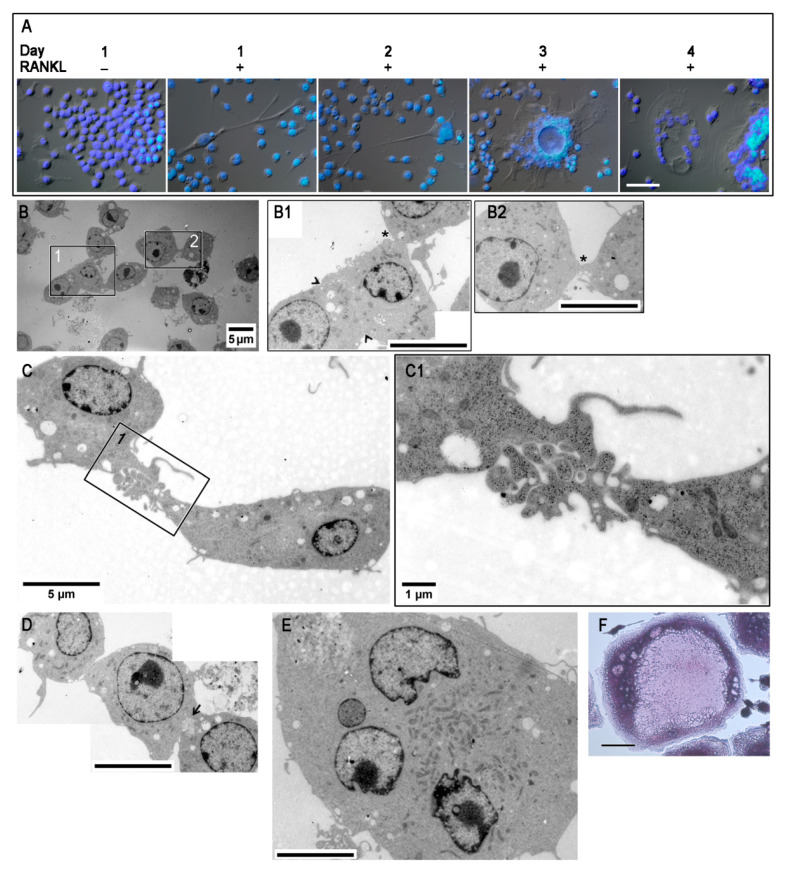
Timing of in vitro osteoclast differentiation. (**A**) Cells cultured in the absence, for 1 day (receptor activator of nuclear factor κ-B ligand (RANKL^−^), or in the presence of RANKL (RANKL^+^) for 1, 2, 3, and 4 days. Merge of brightfield and DAPI staining images, scale bar: 50 µm. (**B**–**E**) TEM images of 2 days RANKL^+^ cells. (**B1**) and (**B2**), 3× magnifications of framed areas in (**B**), showing recognition (*)/adhesion (arrowheads) between mononuclear cells; scale bars 5 µm. (**C1**) 3.5× magnification of framed area in (**C**), showing a large contact area between mononuclear cells. (**D**) Fusing cells through a “fusopode bridge” (arrow); scale bar 5 µm. (**E**) Large multinucleated cell; scale bar 5 µm. (**F**) TRAP staining of 4 days RANKL-stimulated cells. The data shown represent two independent experiments with comparable outcomes. Magnification 40×. Scale bar 50 µm.

**Figure 2 biology-10-00117-f002:**
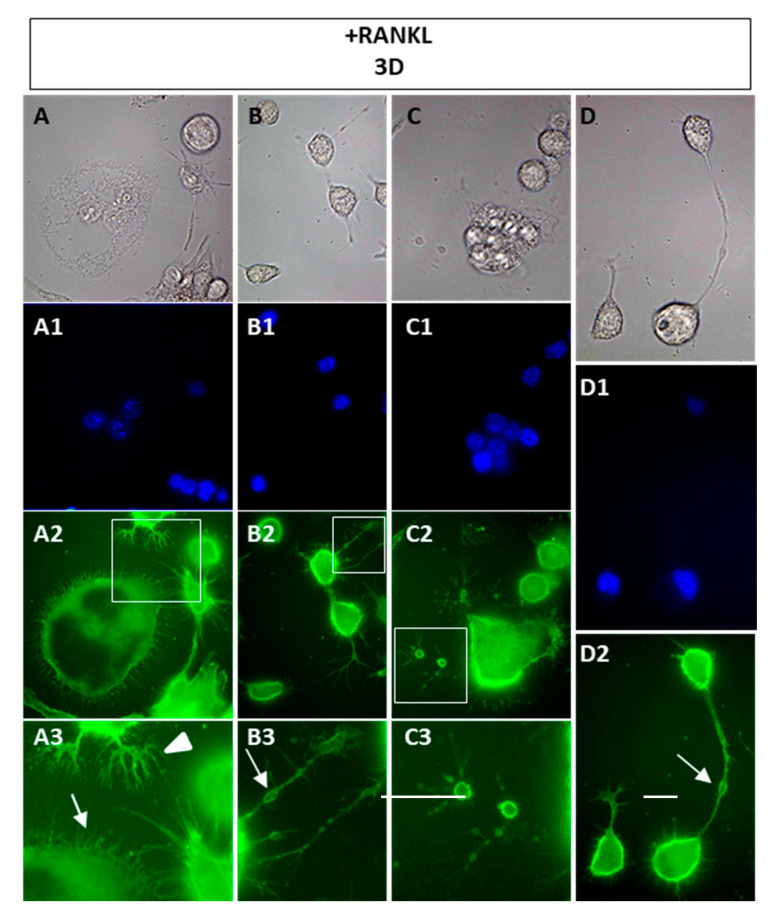
Cellular extensions of 3 days RANKL-stimulated cells. Cells were stained with wheat germ agglutinin (WGA)-FITC and mono or multinucleated cells are shown. (**A**–**D**) brightfield. (**A1**–**D1**) Dapi staining. (**A2**–**D2**) direct immunofluorescence with WGA-FITC. (**A3**–**C3**) 2.5× magnification of **A2**, **B2**, and **C2**, respectively. The data shown represent two independent experiments with comparable outcomes. Magnification 40×. Scale bars of 20 µm.

**Figure 3 biology-10-00117-f003:**
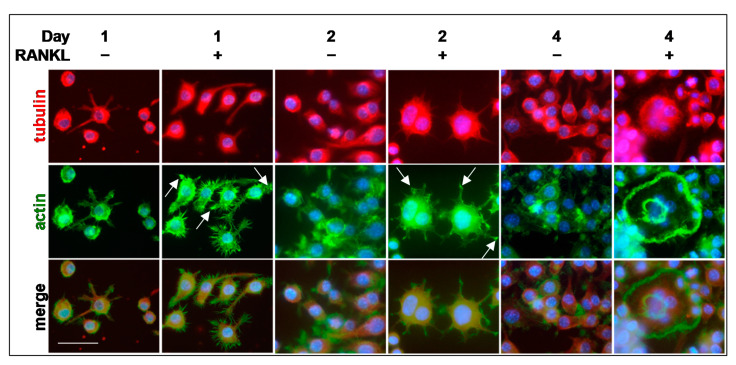
Timing of cytoskeleton organization during OC differentiation. Cells were cultured in absence (RANKL^−^) or in the presence of RANKL (RANKL^+^) for 4 days. Cells were co-stained with anti-tubulin antibody (red), Alexa Fluor-488 phalloidin (green), and DAPI (blue). White arrows, small F-actin clusters. The data shown represent two independent experiments with comparable outcomes. Magnification 40×. Scale bar of 50 µm.

**Figure 4 biology-10-00117-f004:**
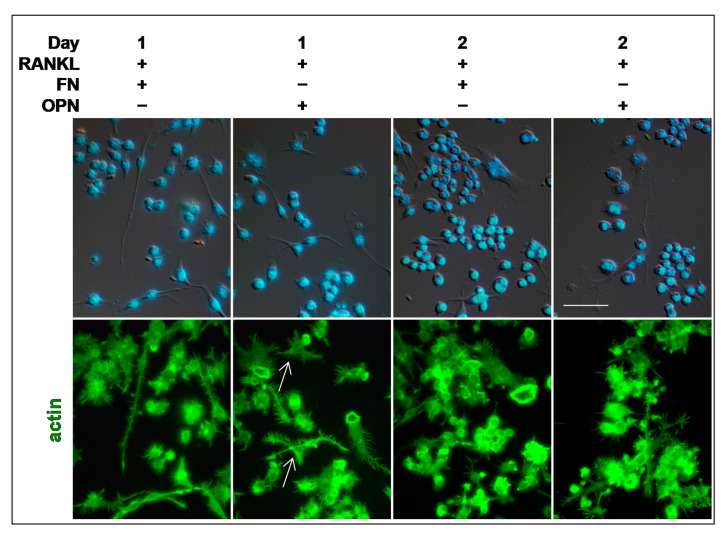
Cytoskeleton organization is modulated by extracellular matrix proteins. Cells were cultured on fibronectin (FN) or osteopontin (OPN) in the presence of RANKL for 1 and 2 days. Merge of brightfield and DAPI staining in the upper row. Alexa Fluor-488 phalloidin (green) in the lower row. White arrows, dendrite-like extensions. The data shown represent two independent experiments with comparable outcomes. Magnification 40×. Scale bar of 50 µm.

**Figure 5 biology-10-00117-f005:**
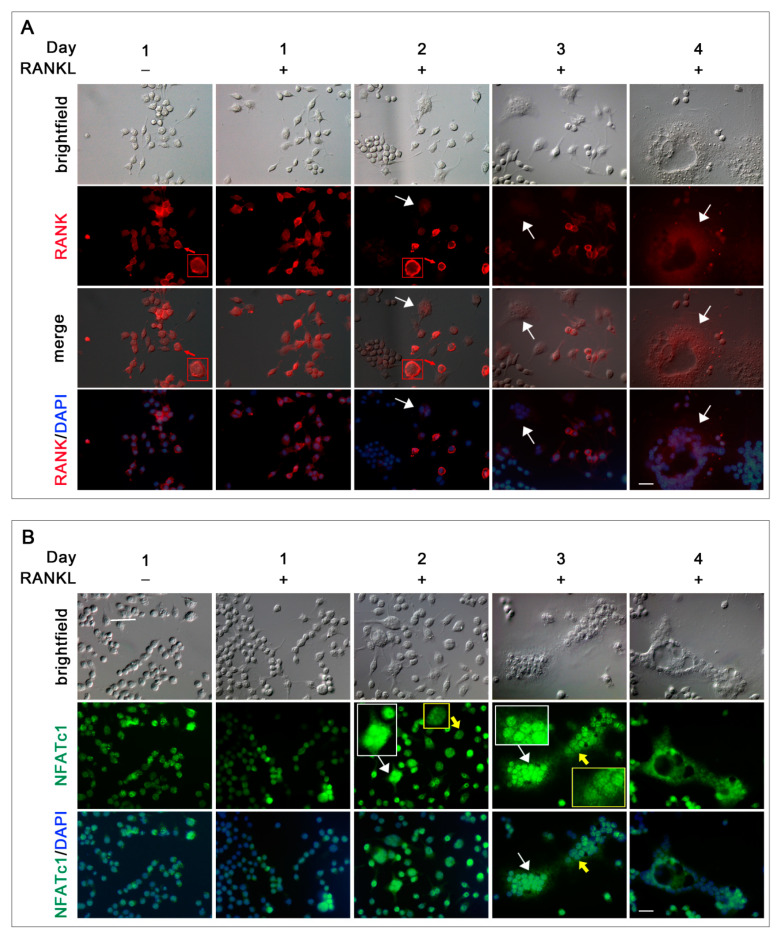
Timing of Receptor Activator of Nuclear factor κ-B Ligand (RANK) and NFATc1 subcellular localization. Cells were cultured in the absence (RANKL^−^) or in the presence of RANKL (RANKL^+^) for 4 days. (**A**) Cells co-stained with anti-RANK antibody (red) and DAPI (blue). White arrows, surface of multinucleated cells. Red frames and arrows indicate 1.5× magnifications of cell membranes RANK-labelled. (**B**) Cells co-stained with anti NFATc1 antibody (green) and DAPI (blue). White frames and arrows indicate 1.5× magnifications of nuclei with high NFATC1 staining intensity. Yellow frames and arrows indicate 1.5× magnifications of nuclei with low NFATC1 staining intensity. The data shown represent two independent experiments with comparable outcomes. Magnification 40×. Scale bar of 50 µm.

**Figure 6 biology-10-00117-f006:**
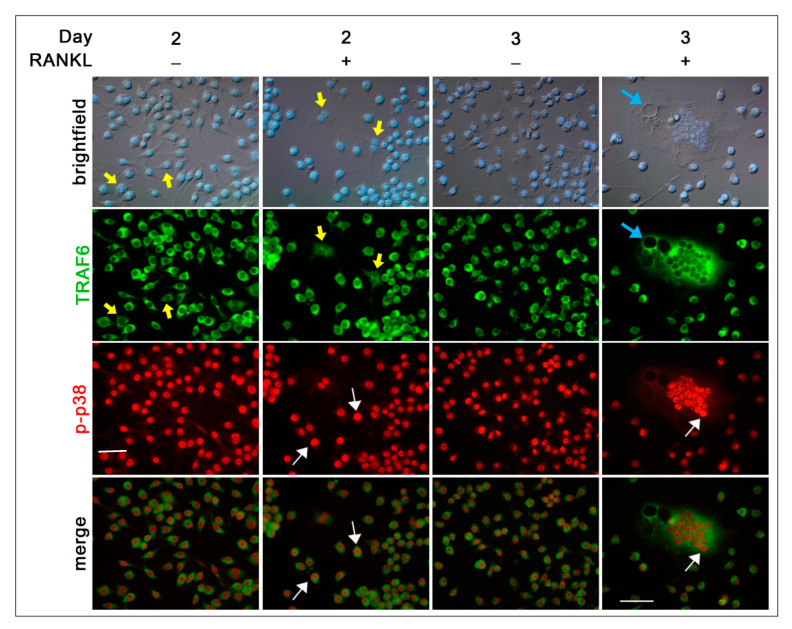
TRAF6 and p-p38 subcellular co-localization. Cells were cultured in the absence (RANKL^−^) or in the presence of RANKL (RANKL^+^) for 2 and 3 days. Cells were stained with anti-TRAF6 (green) and p-p38 (red) antibodies. Yellow arrows, low TRAF6 staining intensity. White arrows, high p-p38 staining intensity. Turquoise arrow, vacuoles in multinucleated cells. The data shown represent two independent experiments with comparable outcomes. Magnification 40×. Scale bar of 50 µm.

**Figure 7 biology-10-00117-f007:**
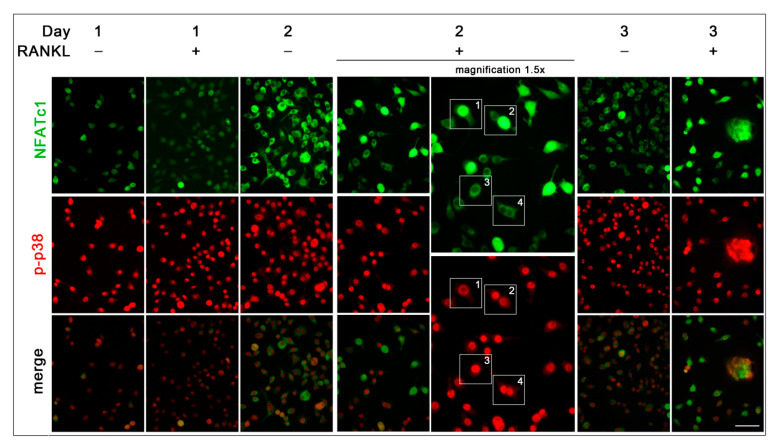
NFATc1 and p-p38 subcellular co-localization. Cells were cultured in the absence (RANKL^−^) or in the presence of RANKL (RANKL^+^) for 1, 2, and 3 days. Cells were stained with anti-NFATc1 (green) and p-p38 (red) antibodies. White frames 1 and 2 indicate 1.5× magnifications of NFATc1 nuclear localization. White frames 3 and 4 indicate 1.5× magnifications of NFATc1 cytoplasmic localization. The data shown represent two independent experiments with comparable outcomes. Magnification 40×. Scale bar of 50 µm.

**Figure 8 biology-10-00117-f008:**
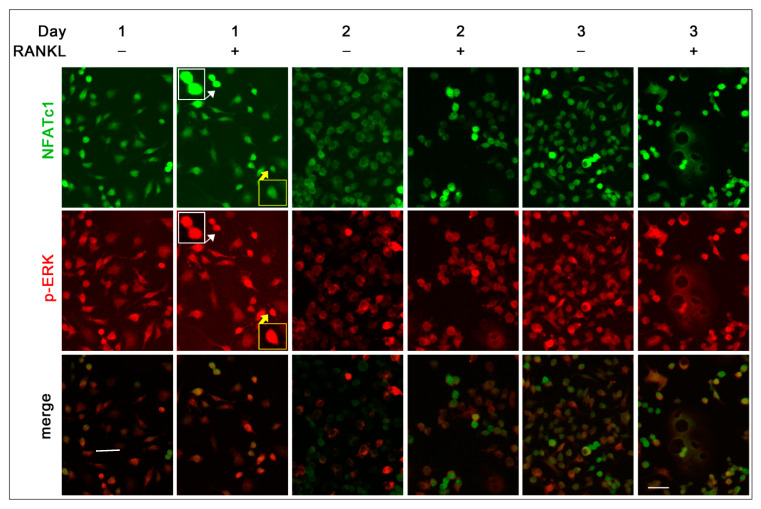
NFATc1 and p-ERK subcellular co-localization. Cells were cultured in the absence (RANKL^−^) or in the presence of RANKL (RANKL^+^) for 1, 2, and 3 days. Cells were stained with anti NFATc1 (green) and p-ERK (red) antibodies. White and yellow frames and arrows indicate 1.5× magnification of p-ERK nuclear localization. White frame indicates 1.5× magnification of NFATc1 and p-ERK co-localization. The data shown represent two independent experiments with comparable outcomes. Magnification 40×. Scale bar of 50 µm.

**Figure 9 biology-10-00117-f009:**
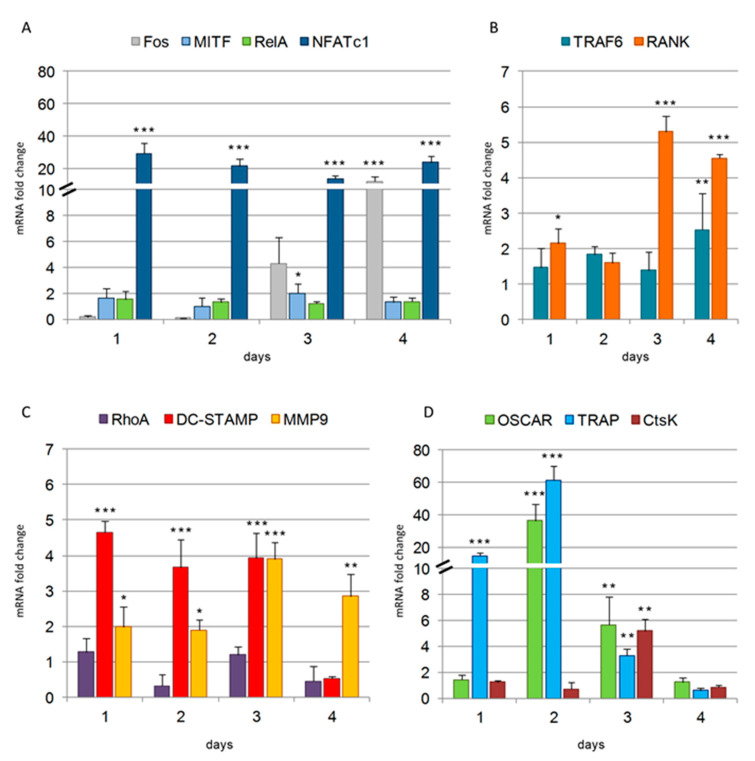
Expression levels of genes during osteoclast (OC) differentiation. (**A**) *Fos*, *MITF*, *RelA*, *NFATc1*; (**B**) *TRAF6*, *RANK*; (**C**) *RhoA*, *DC-STAMP*, *MMP9*; (**D**) *OSCAR*, *TRAP*, *CtsK*; Comparative qPCR analyses of mRNA levels in RAW 264.7 cell cultured in the presence of RANKL for 1, 2, 3, and 4 days. The expression levels of the analyzed genes are presented as relative values of the cells treated with RANKL compared to the cells not treated with RANKL (RANKL^−^) used as controls. Each day sample has its own related RANKL^−^ control (assumed with a value of 1). *GAPDH* was used as a housekeeping gene. Each bar represents the mean of at least three independent qPCR analyses ± SD, using cDNA obtained by two independent OC differentiation experiments. Variations significantly different from the relative controls are indicated by * *p* < 0.05, ** *p* < 0.01, and *** *p* < 0.001.

**Figure 10 biology-10-00117-f010:**
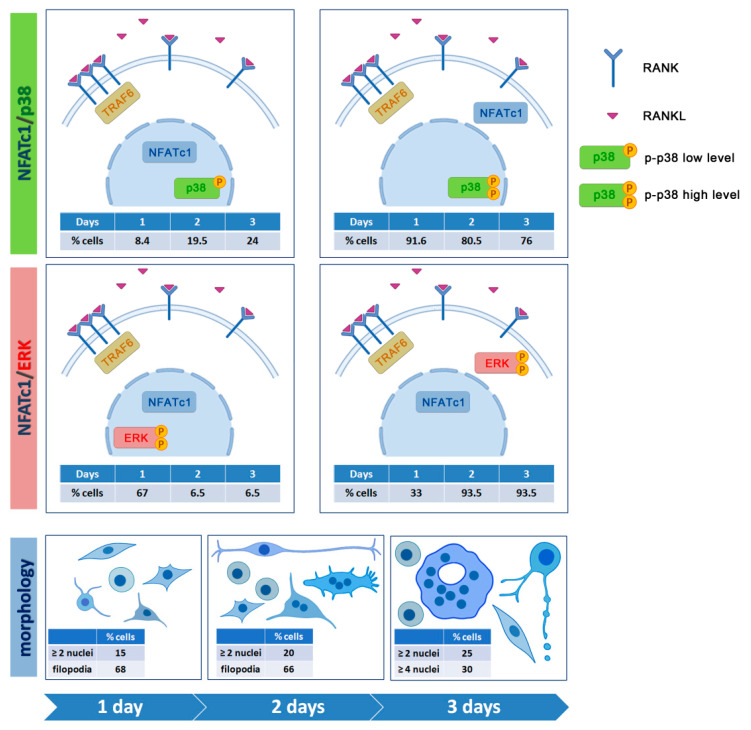
Schematic view of NFATc1, p-p38, p-ERK relationships, and morphologies during OC differentiation. In the upper panels, NFATc1/p-p38 sub-cellular co-localization and the percent of cells (shown in the tables within the panels) at 1, 2, and 3 days are represented. In the intermediate panels, NFATc1/ERK sub-cellular co-localization and the percent of cells (shown in the tables within the panels) at 1, 2, and 3 days are represented. In the lower panels, the various morphologies observed in cell populations at 1, 2, and 3 days are represented.

## Data Availability

Data is contained within the article or Appendix A.

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
