# Peer review of "Osteoclasts Differentiation from Murine RAW 264.7 Cells Stimulated by RANKL: Timing and Behavior"

_biology, 2021, doi:10.3390/biology10020117_

Round 1
Reviewer 1 Report
The manuscript is well written, provides sufficient background information about the study. The study used RAW264.7 cells to understand the pre-Osteoclastogenesis invitro by using RANKL activator, adhesion proteins in the culture for 4days on RAW264.7 cells. They showed various changes in RAW264.7 after treatment with RANKL including multinucleated cells by immunostaining and TEM. The results showed cytoskeleton changes like F actin cluster, dendrite extension, and Filopodia at the invitro level. They further measure gene expression changes related to OC differentiation. I have the following concerns in the manuscript before recommending them for publication.
1) Method section: I suggest the authors include information about the seeding density of RAW 264.7 cells on Day 0 for the Osteroclastogenesis experiment?
2) The scale bar in most of the images were missing, though it's denoted in the Figure legends?
3) Fig1: I would suggest authors include quantitative analysis of multinucleated cells, single nucleated cells from Day 0 to Day 4 with RANKL or without RANKL?
4) Did the authors observed any structure at Filopodia under the TEM? Though Line 366 authors discussed, if possible include TEM picutre.
5) Fig4: Quantitative analysis of images between OPN and FN in terms of cells with or without dendrite extension will be more convincing?
6) Line 238-241: The image resolution is very poor unless a high-resolution image/magnified inlet image, these statement is very strong and needs revision.
7) Fig5: Resolution and quality images of is very poor. Needs high resolution image.
8) Line 269-270: Reference is missing for the statement.
9) Fig6: I would suggest to include brightfield image of corresponding to the fluorescence image ?
10) Line 285-286: Authors should include details about quantification and add details to the method section.
11)Fig 7: Authors need to include high resolution image or maginified inlet image to confirm the co-localization of p-p38 & NFATc1.
12) Did authors performed gene expressionanalysis of FoS,MITF,RelA and NFATc1 from the RAW 264.7 cells before adding RANKL (Day0)?
Author Response
R) 1) Method section: I suggest the authors include information about the seeding density of RAW 264.7 cells on Day 0 for the Osteroclastogenesis experiment?
A) We added this information.
R) 2) The scale bar in most of the images were missing, though it's denoted in the Figure legends?
A) We added all the missing scale bars in all images.
R) 3) Fig1: I would suggest authors include quantitative analysis of multinucleated cells, single nucleated cells from Day 0 to Day 4 with RANKL or without RANKL?
A) We added some quantitative data concerning multinucleated cells (about 15, 20 and 25% of ≥2 nuclei at 1, 2 and 3 days respectively; about 30% of ≥4 nuclei at 3 days), in the Results section and in the Figure 10 showing schematic summary of the results. We want to clarify that the controls for our experiments are the cells cultured in absence of RANKL for the same days as RANKL(+) cells, i.e 1 day RANKL(+) cells are compared with the corresponding 1 day RANKL(-) cells.
R) 4) Did the authors observed any structure at Filopodia under the TEM? Though Line 366 authors discussed, if possible include TEM picutre.
A) In this contest, we did not observe particular structures for filopodia even if they have typical diameter of 200 nm, as it seems that they are not perfectly oriented in the section plane and this type of preparation did not allow the identification of actin or microtubules. However, we added new TEM images (Fig.1C and magnification C1) showing that approaching cells form interlocked cytoplasmic protrusions.
R) 5) Fig4: Quantitative analysis of images between OPN and FN in terms of cells with or without dendrite extension will be more convincing?
A) We added quantitative data in the Results section.
R) 6) Line 238-241: The image resolution is very poor unless a high-resolution image/magnified inlet image, these statement is very strong and needs revision.
A) We improved image resolution and added some framed 1.5x magnifications within Figure 5A and B to better highlight the statements.
R) 7) Fig5: Resolution and quality images of is very poor. Needs high resolution image.
A) In addition to having increased resolution and included magnifications as stated in the previous answer, we added the brightfield images for the figure 5 A and B.
R) 8) Line 269-270: Reference is missing for the statement
A) In the lines indicated, we do not find missing reference. However, we realized that a reference (Fiorino 2016) was missing in the Material and Methods section (line 103). We apologize for the careless error, which we corrected.
R) 9) Fig6: I would suggest to include brightfield image of corresponding to the fluorescence image?
A) We include brightfield images as suggested.
R) 10) Line 285-286: Authors should include details about quantification and add details to the method section.
A) Details about quantification have been added in the Material and Methods section: The percentage of cells showing the nuclear localization of various proteins was calculated as the ratio of nuclei labelled with specific antibodies to the total number of nuclei labelled with DAPI (Molecular Probes, Eugene, OR, USA), counted using the ImageJ software.
R) 11) Fig 7: Authors need to include high resolution image or magnified inlet image to confirm the co-localization of p-p38 & NFATc1.
A) We increased resolution of images and added a 1.5x magnification image, which includes numbered frames to highlight the type of co-localization of p-p38 and NFATc1.
R) 12) Did authors performed gene expression analysis of FoS, MITF, RelA and NFATc1 from the RAW 264.7 cells before adding RANKL (Day0)?
A) No, we did not perform gene expression analyses in cells cultured in absence of RANKL. In particular, the controls for our experiments are always the cells cultured in absence of RANKL for the same days as RANKL(+) cells, i.e 1 day RANKL(+) cells are compared with the corresponding 1 day RANKL(-) cells, 2 days RANKL(+) cells are compared with the corresponding 2 days RANKL(-) cells and so on. To avoid confusion, we have added a sentence in the legend of Figure 9: The expression levels of the analyzed genes are presented as relative values of the cells treated with RANKL compared to the cells not treated with RANKL (RANKL-) used as controls. Each day sample has its own related RANKL- control (assumed with a value of 1).
Reviewer 2 Report
The authors report on in vitro oestoclastogenesis through RANKL stimulation over a 4 day period with analysis of morphological and cytoskeletal changes, changes in subcellular localisation of key molecular players (RANKL, NFATc1, p38 etc) by IF and also mRNA gene expression analysis of TFs and signalling molecules involved in the process.
In general, the manuscript is well written but with some minor errors. The data presented is generally of good quality.
Points to address:-
What are RAW264.7 cells? – this should be made clear in the abstract at first mention (line 13) to assist understanding by a broad scientific readership.
Why were murine rather than human cells used? Do you think the timing and behaviour may be different in different species and cell models?
Some minor errors of English – please proofread carefully, e.g. genes expression rather than gene expression; proteins distribution rather than protein distribution (lines 15,16, line 61)
Don’t use 2D or 3D to represent 2 days or 3 days – as this can confuse with 2- and 3-dimensional (abstract). Needs correcting throughout the manuscript.
There is an assumption of knowledge that readers will know what WGA-FITC is (abstract, line 19). Please address.
Line 40 – replace in with ‘into’
Line 56 – replace transduced with transduce
Line 65- capital letters for names (ATCC)
Line 69 – at what cell density were cells seeded/cm2 – or if suspension cells as the text suggests cells/ml? clarity needed
Line 73 – was followed, over 4 days
Replace osteoclasts differentiation with oestoclast differentiation throughout the manuscript
Figures 1, 2, 3, 4 and throughout – scale bars are missing.
Line 200 – delete ‘plasma membranes’
Line 213 – 4-????
For Fig. 5 – please show brightfield images also so location of the cell membrane/cytoplasm can be more clearly discerned. To support the claims made in terms of changes in subcellular localisation of RANKL and NFATC this would be assisted by analysis additionally by immunoblotting of subcellular fractions.
The discussion would benefit greatly from a summary schematic bringing together the key findings of the different analyses and of the paper as a whole - including the interesting migrasomes.
Author Response
R) 1) What are RAW264.7 cells? this should be made clear in the abstract at first mention (line 13) to assist understanding by a broad scientific readership.
A) Now we included the cell line details in the abstract.
R) 2) Why were murine rather than human cells used? Do you think the timing and behaviour may be different in different species and cell models?
A) We used RAW 264.7 cell line because they are easy to use and in a time of 4 days they differentiate into mature osteoclasts. We know very well this model and we studied it since many years. In addition, mouse cells are the most studied compared to human cells, also because ex vivo experiments can be performed by taking osteoclast precursors from the bone marrow. In the literature, it is shown that timing and behaviour of osteoclasts differentiation are different in different species.
R) 3) Some minor errors of English – please proofread carefully, e.g. genes expression rather than gene expression; proteins distribution rather than protein distribution (lines 15,16, line 61)
A) English editing was performed on the entire text.
R) 4) Don’t use 2D or 3D to represent 2 days or 3 days – as this can confuse with 2- and 3-dimensional (abstract). Needs correcting throughout the manuscript.
A) We change all the terms in the manuscript
R) 5) There is an assumption of knowledge that readers will know what WGA-FITC is (abstract, line 19). Please address.
A) Now, we wrote the full name for WGA-FITC is in the abstract.
R) 6) Line 40 – replace in with ‘into’
A) We did not find the word to replace in line 40
R) 7) Line 56 – replace transduced with transduce; 8) Line 65- capital letters for names (ATCC); 10) Line 73 – was followed, over 4 days; 13) Line 200 – delete ‘plasma membranes’
A) We have made all the suggested changes
R) 9) Line 69 – at what cell density were cells seeded/cm2 – or if suspension cells as the text suggests cells/ml? clarity needed
A) Now, we specify the cell density for all the experiments
R) 11) Replace osteoclasts differentiation with oestoclast differentiation throughout the manuscript????
A) We have made all the suggested changes.
R) 12) Figures 1, 2, 3, 4 and throughout – scale bars are missing
A) Now, we added the scale bars
R) 14) Line 213 – 4-????
A) We wrote more clearly what we mean
R) 15) For Fig. 5 – please show brightfield images also so location of the cell membrane/cytoplasm can be more clearly discerned. To support the claims made in terms of changes in subcellular localisation of RANKL and NFATC this would be assisted by analysis additionally by immunoblotting of subcellular fractions.
A) We added the brightfield images and some framed 1.5x magnifications within Figure 5A and B to better highlight the statements. The reviewer suggestion is right. Unfortunately, in this period due to the Covid 19, we have very limited access to the laboratory and we are unable to do this experiment for this manuscript.
R) 16) The discussion would benefit greatly from a summary schematic bringing together the key findings of the different analyses and of the paper as a whole - including the interesting migrasomes.
A) Now, we added a schematic summary, which we hope will satisfies the request of the reviewer.
Round 2
Reviewer 1 Report
The current revised version of the manuscript is addressed all my concerns raised in the previous version. I recommend this version for publication.